# Fibroblast Growth Factor 21 Stimulates Pancreatic Islet Autophagy via Inhibition of AMPK-mTOR Signaling

**DOI:** 10.3390/ijms20102517

**Published:** 2019-05-22

**Authors:** Sam Tsz Wai Cheng, Stephen Yu Ting Li, Po Sing Leung

**Affiliations:** School of Biomedical Sciences, Faculty of Medicine, Chinese University of Hong Kong, Hong Kong, China; samsn_n@hotmail.com (S.T.W.C.); stephenliyuting_1992@yahoo.com.hk (S.Y.T.L.)

**Keywords:** diabetes mellitus, islet autophagy, FGF21, CVX343

## Abstract

Background: Islet autophagy plays a role in glucose/lipid metabolism in type 2 diabetes mellitus. Meanwhile, fibroblast growth factor 21 (FGF21) has been found to regulate insulin sensitivity and glucose homeostasis. Whether FGF21 induces islet autophagy, remains to be elucidated. This study aimed to explore the physiological roles and signaling pathways involved in FGF21-stimulated islet autophagy under glucolipotoxic conditions. Methods: C57/BL6J mice were fed a standard diet or high-fat diet (HFD) for 12 weeks, and islets were isolated from normal and *FGF21* knockout (KO) mice. Isolated islets and INS-1E cells were exposed to normal and high-concentration glucose and palmitic acid with/without FGF21 or AMPK inhibitor compound C. Real-time PCR, Western blot and immunohistochemistry/transmission electron microscopy were performed for the expression of targeted genes/proteins. Results: HFD-treated mice showed increases in fasting plasma glucose, body weight and impaired glucose tolerance; islet protein expression of FGF21 was induced after HFD treatment. Protein expression levels of FGF21 and LC3-II (autophagy marker) were induced in mouse islets treated with high concentrations of palmitic acid and glucose, while phosphorylation of AMPK was reduced, compared with controls. In addition, induction of LC3-II protein expression was reduced in islets isolated from *FGF21* KO mice. Furthermore, exogenous administration of FGF21 diminished phosphorylation of AMPK and stimulated protein expression of LC3-II. Consistently, compound C significantly induced increased expression of LC3-II protein. Conclusions: Our data indicate that glucolipotoxicity-induced FGF21 activation mediates islet autophagy via AMPK inhibition, and further consolidate the evidence for the FGF21/analog being a pharmacotherapeutic target for obesity and its related T2DM.

## 1. Introduction

It is estimated over 400 million people are suffering from diabetes worldwide, with type 2 diabetes mellitus (T2DM) being the predominant form, which accounts for >90% of all cases [1]. T2DM develops when insulin resistance increases in insulin-target organs, followed by impaired insulin secretion and reduced β-cell mass [2]. Obesity is a well-known risk factor for T2DM progression, and obesity-induced glucolipotoxicity (i.e., hyperglycemia and hyperlipidemia) inflicts oxidative stress, thus leading to impaired islet glucose-stimulated insulin secretion (GSIS) and apoptosis [3].

Fibroblast growth factor (FGF) 21 is a potent metabolic regulator, expressed predominantly in the liver, but also in adipose tissue and the pancreas [4]. Pharmacological studies have demonstrated that FGF21 improves insulin sensitivity, thereby countering the development of metabolic diseases, including obesity and T2DM [4]. 

Circulating FGF21 levels are elevated in rodents and humans with obesity and T2DM [5,6]. In fact, FGF21 levels were found to correlate inversely with whole-body insulin sensitivity, and to correlate directly with the hepatic insulin resistance index and glycemia [5]. In addition, exposure of islets and INS-1E cell β-cells to diabetic conditions attenuates FGF21 responsivity, as evidenced by reductions in extracellular mitogen-activated protein kinase 1 and 2 (ERK1/2) phosphorylation, Protein kinase B (PKB) signaling, and FGF21 target gene expression [6,7]. Interestingly, people who are obese or diabetic exhibit FGF21 resistance [6]. Moreover, high glucose conditions impaired FGF21’s action in mouse islets via a suppression of β-klotho (KLB), which was ameliorated by rosiglitazone, an agonist for peroxisome proliferator-activated receptor-γ [8]. Although extensive studies confirmed the beneficial role of FGF21 in insulin sensitivity and glucose homeostasis in liver and adipose tissues, less attention has been paid to glucolipotoxicity-induced pancreatic β-cell dysfunction [7]. In this regard, *FGF21* knockout (KO) mice exhibited abnormal islet morphology, islet dysfunction, and impaired GSIS due to a dysregulation of growth hormone signaling [9]. These findings point toward FGF21 being functional in islets in addition to insulin-target organs. 

Macroautophagy (herein referred to as autophagy) is a regulated catabolic process used by eukaryotic cells to degrade aggregated or damaged proteins and organelles [10,11]. Autophagy plays a key role in nutrient sensing and cell growth [12]. The observation of impaired autophagy in association with β-cell lipotoxicity and dysfunction suggests that autophagy may be critical for the regulation of glucose and lipid homeostasis [13]. Autophagy may be particularly important in β-cells because of their especially high protein (i.e., insulin) synthesis levels [14]. Increased β-cell autophagic flux, as reflected by the formation of autophagosomes and the induction of autophagy-related genes (e.g., *LC3*), has been observed in C57BL/6 mice fed a 60% high-fat diet (HFD) and in β-cells treated with palmitate [14,15]. In addition, β-cell-specific autophagy-deficient mice could exhibit progressive β-cell degeneration, hypoinsulinemia, and hyperglycemia [16]. The magnitude of β-cell degeneration in these mice became more significant when they were fed an HFD, indicative of being hyper-susceptible to lipotoxicity [17]. Observation of elevated serum levels of free fatty acids (FFAs) in association with insulin resistance is consistent with the possibility that FFA-mediated induction of β-cell autophagy may alter β-cell function in obesity-related diabetes [18,19,20]. If confirmed, β-cell survival and function may be improved by pharmacological stimulation of autophagy [21,22]. The relationship between pancreatic β-cells and autophagy remains to be fully clarified. 

Autophagy is tightly regulated by two signaling molecules, namely mTOR (mechanistic target of rapamycin) and AMPK (5’ AMP-activated protein kinase) [23]. The mTOR interacts with various binding proteins to form at least two functionally distinct complexes, namely mTOR1 and 2 [23]. Any mTOR2 activity is suppressed under starvation or FFA exposure, a key trigger of autophagy induction in eukaryotes [23,24]. Activation of AMPK requires kinase phosphorylation of a conserved threonine residue (T172) in the activation loop of its catalytic α-subunit [25,26,27]. The roles of AMPK as a major metabolic energy sensor and regulator are well known [25]. The anti-diabetic actions of metformin have been shown to be AMPK-dependent [18]. Interestingly, administration of recombinant FGF21 protein to alcohol-treated mice and HepG2 cells enhanced phosphorylated AMPK levels in both the mouse liver cells and the in vitro liver cells [28]. Additionally, FGF21 treatment increased AMPK phosphorylation in 3T3-L1 adipocytes, white adipose tissues and human differentiated adipocytes [29]. Conversely, chronic exposure of β-cells to palmitate has been shown to inhibit AMPKα activity, consistent with mTOR inhibition by long-term FFA exposure [30,31,32]. These findings point to AMPK and mTOR2 being able to interact in autophagic induction, perhaps through direct activation of ULK1 [33]. On the other hand, it has been reported that adiponectin, a downstream effector of *FGF21*, can stimulate autophagy, reduce oxidative stress in skeletal muscles and, ultimately, enhance insulin sensitivity in mice fed an HFD [34]. Meanwhile, fenofibrate has been shown to increase cardiocyte autophagy via the FGF21 pathway in T1DM mice [35]. Additionally, FGF21 itself has been reported to promote the expression of several autophagy-related proteins in both obese mice and fat-loaded hepatocytes, as well as to increase autophagosome and lysosome numbers in liver cells [36]. 

The above discussed convergence of findings suggests that FGF21 plays a positive role in activating autophagy in various cells and tissues. We thus hypothesize that FGF21 may have a regulatory role in islet autophagy-mediated islet function and survival via AMPK-mTOR signaling, that is protective against glucolipotoxicity-induced islet dysfunction in obesity-associated T2DM. To test this hypothesis, we conducted a series of experiments to investigate the mechanism-of-action involved in FGF21-stimulated autophagy in pancreatic islets isolated from HFD-induced diabetic mice and *FGF21* KO mice, as well as in isolated islets and insulinoma INS-1E cells subjected to pharmacological manipulations.

## 2. Results

### 2.1. FGF21 Analog CVX343 Improves Glucose Homeostasis and Body Weight in HFD-Induced T2DM Mice

Compared to control mice fed a standard diet, HFD feeding increased the body weight of mice significantly; however, the HFD effect on weight gain was attenuated significantly by high-dose CVX343 (10 mg/kg body weight (BW)), while a nonsignificant attenuation trend was observed with low-dose CVX343 (3 mg/kg BW) (Figure 1A). Meanwhile, both CVX343 treatments resulted in improved fasting blood glucose levels (Figure 1B) as well as improved glucose intolerance (Figure 1C) in HFD-treated mice relative to untreated controls, as evidenced by IPGTT results. In addition, high-dose CVX343 ameliorated insulin resistance significantly in HFD-treated mice, as evidenced by ITT results (Figure 1D).

### 2.2. In Vivo HFD-Induced T2DM and Ex Vivo High-Glucose/High-PA Treatment Upregulate Protein Expression of FGF21 and LC3-II in Pancreatic Islets

HFD feeding for 12 weeks to induce diabetes increased islet protein expression of FGF21 (Figure 2A) and the autophagy marker LC3-II (Figure 2B) relative to the chow diet, as demonstrated by Western blot analysis. In addition, LC3-II expression was increased in a time-dependent manner in isolated islets cultured ex vivo in high-glucose/high-PA conditions (Figure 2C), whereas protein levels of FGF21 were also increased significantly after 24-h exposure to high-glucose/high-PA conditions (Figure 2D) relative to the control, as demonstrated by Western blot analysis.

### 2.3. Pancreatic Islets Isolated from FGF21 KO Mice and INS-1E β-Cells with FGF21 Knockdown Display Diminished Autophagy Induction

Real-time PCR and Western blot analyses showed that siRNA #3 down-regulated FGF21 in INS-1E β-cells (Figure 3A,B). LC3-II expressions were also found in *FGF21* knocked down INS-1E cells subjected to exposure to high-glucose/high-PA conditions, as evidenced by Western blot analyses (Figure 3C). RT-PCR showed that *FGF21* gene in pancreatic islets from *FGF21* KO mice was successfully knocked out (Figure 3D). In corroboration with the INS-1E cell, when cultured in high-glucose/high-PA conditions, consistent results on *FGF21* KO mouse islets had reduced LC3-II protein expression compared to islets isolated from their respective WT mice (Figure 3E). 

To further confirm the functional role of FGF21-mediated autophagy induction, we performed immunocytochemistry and transmission electron microscopy (TEM) examinations. Results from immunofluorescent labeling revealed decreased cytoplasmic LC3-II expression in high-glucose/high-PA-treated INS-1E cells with *FGF21* knockdown, compared to non-treated control cells (Figure 4A). On the other hand, TEM revealed that autophagy induction was impaired in *FGF21* KO islets under diabetic conditions (Figure 4B), as evidenced by the reduced formation of isolation membranes (Figure 4C) and autophagosomes (Figure 4D), in relation to their respective control.

### 2.4. In Vivo HFD, Ex Vivo High-Glucose/High-PA, and Exogenous Recombinant FGF21 Treatments Each Reduced AMPK Phosphorylation in Pancreatic Islets

Immunoblot analysis revealed that the protein expression of AMPK phosphorylation was suppressed in pancreatic islets isolated from mice fed a 12-week HFD (Figure 5A), relative to those from mice fed a standard chow diet. Similarly, INS-1E β-cells exposed to high-glucose/high-PA conditions for 24 h was able to reduce AMPK phosphorylation (Figure 5B). Furthermore, exogenous administration of FGF21 recombinant protein increased protein expression of islet FGF21 in a time-dependent manner over respective time points (Figure 5C). Meanwhile, AMPK phosphorylation was suppressed transiently in isolated islets treated with exogenous FGF21 recombinant protein (Figure 5D).

### 2.5. Compound C Suppressed mTOR Phosphorylation but Increased LC3-II Expression

To further explore the potential interaction between AMPK and mTOR-mediated autophagic pathways, we sought to employ compound C, which is a potent AMPK blocker. Our results showed that incubation with the potent AMPK blocker compound C (20 μM) reduced mTOR phosphorylation significantly (Figure 6A,B), while increasing LC3-II protein expression significantly (Figure 6A,C) in relation to the control, as evidenced by Western blot analysis.

## 3. Discussion

In this study, we demonstrated that weight gain, glucose intolerance and insulin resistance in HFD-fed mice could be reversed by an application of the FGF21 mimetic CVX343. This observation was previously supported in diet-induced obese mice, showing that weekly doses of CVX343 were able to lower body weight, blood glucose and lipids levels whereas, in db/db mice, CVX343 improved glucose intolerance and increased pancreatic β-cell mass [37]. Besides, FGF21 expression was upregulated in pancreatic islets of HFD-fed mice, as well as in high-glucose/high-PA-treated isolated islets. Exposure to high-glucose/high-PA conditions increased the expression of LC3-II, as well as the formation of isolation membranes and autophagosome in islets isolated from WT mice; these demonstrated effects were consistent with glucolipotoxicity-induced autophagy, but not in islets from *FGF21* KO mice. Exogenous recombinant FGF21 suppressed AMPK phosphorylation and increased LC3-II protein expression, while AMPK inhibition attenuated mTOR phosphorylation and upregulated LC3-II protein expression. All these data point to the idea that *FGF21* activation under glucolipotoxicity may induce autophagic flux via inhibition of AMPK-mTOR signaling in islets, and that *FGF21* may be an important mediator of islet β-cell functions and pathophysiology related to obesity and T2DM. 

Our findings on exogenous FGF21 administration-induced enhanced expression of FGF21 in a time-dependent manner indicate that our observation of increased FGF21 expression in islets from HFD mice may be a result of increased circulating FGF21 levels under diabetic conditions. 

Circulating FGF21 levels have been previously reported to be elevated in T2DM animal models and in human patients [5,38,39]. Regardless, an increase in the plasma level of FGF21 may serve as a potential biomarker for early-stage metabolic disorders. The question of whether such elevations in FGF21 are due to a compensatory response remains to be further investigated [4]. 

Our in vivo data showing that CVX343 lowered body weight, improved islet function and potentiated insulin secretion in response to a glucose challenge, as well as enhanced insulin sensitivity in HFD group mice, are in corroboration with previously reported in vivo effects of recombinant FGF21 protein on diabetic monkeys and rodents [40,41,42]. If confirmed, CVX343 may act as a novel anti-diabetic agent. Because our high-dose CVX343 group mice exhibited weight loss, it will be important to determine whether such body weight loss is related to the promotion either by physiological processes (e.g., reduced food intake or increased fat catabolism), or by toxicity. 

The present study findings are complementary to prior work, showing that FGF21 administration can improve glucose tolerance and insulin sensitivity in diabetic rodents [40,41,42,43] as well as a prior study showing that transgenic overexpression of *FGF21* is protective against HFD-induced obesity in mice [3]. Moreover, mice lacking *FGF21* or autophagy-related genes display impaired insulin secretion, abnormal islet cell growth, and greater susceptibility to T2DM [9,18,44]. The association of insulin resistance and islet autophagy with elevated serum levels of FFAs suggests that FFAs may play a compensatory, protective role in islets against obesity-associated diabetes pathology [16,17,18]. Emerging studies have accumulated that FGF21 is involved in the induction of autophagy in heart, liver, and skeletal muscle tissues [34,35,36]. As such, dysregulation of islet autophagy represents a plausible mechanism of glucolipotoxicity-mediated islet dysfunction. 

It is well known that AMPK plays important roles in the control of cell growth and metabolism reprogramming, and AMPK was linked recently to cellular processes, such as autophagy [45]. FGF21 has been shown to regulate AMPK phosphorylation in tissues other than pancreatic islets; for example, FGF21 treatment reversed alcoholic fatty liver and hepatic steatosis via activation of the hepatic AMPK-SIRT1 pathway [28,46]. Likewise, FGF21 regulates energy metabolism via activation of the AMPK-SIRT1-PGC-1α pathway in adipocytes [29]. However, FGF21 had an opposite effect on AMPK activities in our study, that is, FGF21 down-regulated AMPK phosphorylation in pancreatic islets. Therefore, it is plausible to postulate that FGF21 may mediate a different metabolic role in islets than in the liver and adipose, which are the primary targets of pancreatic hormones (insulin and glucagon) in the control of blood glucose homeostasis. Since both HFD feeding and ex vivo high-glucose/high-PA treatment induced FGF21 expression and suppressed AMPK phosphorylation, we deduce that islet FGF21 may alter AMPK activities during the pathogenesis of T2DM. 

Given mTOR reduction in the induction of autophagy [24], it is noteworthy that AMPK inhibition by high fat treatment or compound C in this study decreased mTOR phosphorylation and induced signs of autophagy in pancreatic islets. The result was supported by a previous finding that compound C induces autophagy in cancer cells through blockade of mTOR pathways [47]. Our study is the first to report that FGF21 may regulate mTOR activity and, in turn, autophagy, via the inhibition of AMPK phosphorylation. Our data are in agreement with a prior study showing that hyperactivation of mTOR led to autophagy impairment in β-cells, thus subsequently contributing to β-cell failure [48]. 

Although considerable research has pointed to beneficial effects of autophagy on promoting β-cell survival in T2DM, it is possible that activation of autophagy may, in some circumstances, lead to β-cell death [49]. Indeed, down-regulation of pdx1 in mice within MIN6 β-cells can also trigger islet autophagy and cell death, whereas pharmacological inhibition of autophagy prevents autophagic cell death [50]. Moreover, several components of the extrinsic apoptotic pathway, such as tumor necrosis factor and ceramide, have been reported to induce autophagy [51]. Given that autophagy can either protect cells or promote cell death, depending on the cellular and environmental context [52], further investigation is warranted to elucidate the connection between cell survival and apoptosis in pancreatic β-cells undergoing autophagy.

## 4. Materials and Methods

### 4.1. Animal Models

Male C57BL/6J mice (6 weeks of age) were supplied by the Laboratory Animal Service Center of the Chinese University of Hong Kong. Prior to this study, the mice had been housed at 22 ± 2 °C with a 12-h light/dark cycle and provided with an ad labitum rodent chow diet. From 6 weeks old onwards, they received either standard rodent chow or a high-fat diet (HFD) (60% kCal from fat; Teklad, Harlan Laboratories, Madison, WI, USA) for 12 weeks to establish an HFD-induced type 2 diabetes mellitus (T2DM) model. Male fibroblast growth factor 21 (*FGF21*)-knockout (KO) and wild-type (WT) mice of C57BL/6J origin were employed as we described previously [9]. All experimental procedures were approved by the Animal Experimentation Ethics Committee of the Chinese University of Hong Kong (Ref. # 15/054/GRF/4-A, 10/03/2015).

### 4.2. FGF21 Analog Supplementation

Male C57BL/6J mice were assigned randomly to one of the following groups: Low-dose CVX343 (3 mg/kg body weight (BW)); high-dose CVX343 (10 mg/kg BW); or non-CVX343 treated control (200 μl of saline). After 6 weeks of HFD feeding (week 7–12), the FGF21 analog CVX343 (PF-05231023, a gift from Pfizer) was administered weekly for 6 weeks by intraperitoneal injection.

### 4.3. Body Weight and In Vivo Glucose Homeostasis

Once a week during the CVX343 treatment period, the mice were weighed, and their fasting blood glucose levels (after 6-h fasting) were measured in blood samples drawn from the tail vein by a glucometer (Bayer Corporation, Leverkusen, Germany). Intraperitoneal glucose tolerance tests (IPGTTs) were administered after a 6-h fast and subsequent glucose challenge (1 g/kg BW); blood glucose was measured 0, 15, 30, 60, 90, and 120 min thereafter. For insulin tolerance tests (ITTs), mice were injected with insulin (0.5 U/kg BW; Eli Lilly and Company, Indianapolis, IN, USA) after a 4-h fasting; blood glucose levels were subsequently measured. Areas under the curve (AUCs) for BW and blood glucose levels were calculated.

### 4.4. Pancreatic Islet Isolation and Treatments

Collagenase P (Roche, Mannheim, Germany) was injected intraductally into harvested pancreata in order to isolate intact pancreatic islets as described previously [8,9]. Isolated pancreatic islets were incubated overnight before treatment with 5.6 or 28 mM d-glucose (Sigma-Aldrich, St. Loise, MO, USA) and 0 or 0.5 mM palmitic acid (PA) in the presence of 100 nM FGF21 recombinant protein (AIS, HKU) and/or 20 μM AMPK blocker Compound C (Sigma-Aldrich) for the designated time periods.

### 4.5. INS-1E Cell Culture and Treatments

Rat insulinoma INS-1E cells, which is a gift from Dr. Pierre Maechler, [53] were cultured in a humidified chamber with 5% CO_2_ in RPMI 1640 medium (11.2 mM glucose), supplemented with 10% fetal bovine serum, 1 mM sodium pyruvate, 50 μM 2-mercaptoethanol, 10 mM HEPES, 100 U/mL penicillin, and 100 μg/mL streptomycin (all from Invitrogen, Waltham, MA, USA). Cultures were passaged once a week by gentle trypsinization. INS-1E cells were treated with glucose (11.2 or 28 mM), PA (0 or 0.5 mM), and FGF21 recombinant protein (100 nM) for the indicated time.

### 4.6. Knockdown of FGF21 Transcription

Gene expression of *FGF21* was suppressed with small interfering RNAs (siRNAs) for rat *Fgf21* (constructed by Life Technologies, Hong Kong, China). The siRNA negative control oligonucleotides (siRNA-NC) or siRNA-*FGF21* oligonucleotides (sequences in Table 1) were transfected into INS-1E cells, by lipofectamine RNAi Max transfection reagent (Invitrogen) for 48 h, according to the manufacturer’s protocols.

### 4.7. Reverse Transcriptase (RT)-PCR and Real-Time PCR Analysis

Total RNA from pancreatic islets and INS-1E cells was extracted by TRIzol reagent (Invitrogen), according to the manufacturer’s instructions. Reverse transcription of first-strand cDNA was performed with a PrimeScript reverse transcriptase master mix kit (Takara Bio Inc., Kusatsu, Japan). Gene expression was measured by conventional PCR or real-time PCR, wherein cDNA samples were mixed with SYBRgreen QPCR master mix (Applied Biosystems, Waltham, MA, USA) and specific primers (Table 1). The fold change of mRNA expression relative to the control group was obtained using the 2^−ΔΔCt^ method, and normalized to glyceraldehyde-3-hosphate dehydrogenase (*GAPDH*; in mouse pancreatic islets) or *β-actin* (in rat INS-1E cells).

### 4.8. Western Blot Analysis

Islet and cell proteins were extracted with CytoBuster Protein Extraction Reagent (Novagen, Darmstadt, Germany). Extracted proteins were separated by 8–12% sodium dodecyl sulfate-polyacrylamide gel electrophoresis, and transferred to nitrocellulose membranes (Bio-rad, Heidemannstraße, Germany), which were blocked with 5% milk and then probed overnight with anti-FGF21 (Abcam, Cambridge, UK), anti-LC3 (Novus, St. Charles, MO, USA), anti-phospho-AMPKα, anti-AMPKα, anti-phospho-ERK1/2, anti-ERK1/2, anti-phosphomTOR, anti-mTOR (Cell Signaling Technology, Boston, MA, USA), or anti-β-actin (Santa Cruz Biotechnology, Santa Cruz, CA, USA) primary antibodies at room temperature. Horseradish peroxide-conjugated secondary antibodies were incubated at RT with the membranes for 2 h after washing with Phosphate buffered saline with Tween-20. Labeled protein bands were visualized on autoradiography films (Fuji Film, Tokyo, Japan) following application of ECL detection reagent (GE Healthcare, Chicago, IL, USA). The protein bands were quantitated in ImageJ software (National Institutes of Health, Bethesda, MD, USA) and normalized to β-actin. The primary and secondary antibodies used are listed in Table 2.

### 4.9. Immunohistochemistry

Pancreatic islets and rat insulinoma INS-1E β-cells were embedded in O.C.T. compound (Sakura, Tokyo, Japan) and frozen. Cryostat sections (6 μm thick) were cut, mounted, and blocked in 2% bovine serum albumin (Sigma-Aldrich,) with 0.1% Triton X-100 (Sigma-Aldrich) for 30 min at RT. For the autophagy analyses, sections were probed with polyclonal rabbit anti-LC3 antibody (Novus) and guinea pig polyclonal anti-insulin antibody (Life Technologies, Carlsbad, CA, USA) and then incubated for 1 h at RT with Alexa Fluor^®^ 568 donkey anti-rabbit antibody and Alexa Fluor^®^ 488 goat anti-guinea pig (both from Life Technologies). DAPI nuclear counterstain was applied, and then the sections were washed three times with phosphate buffer saline, mounted with VectaShield mounting medium (Vector Laboratories, Burlingame, CA, USA), and observed under a fluorescent microscope (Olympus FV1200 Confocal system with a motorized stage and SIM scanner). Antibodies and dilutions used are listed in Table 2.

### 4.10. Detection of Autophagosome Formation 

Mouse islets were rinsed with 0.1 M Sorensen phosphate buffer (pH 7.2) after being cultured. Specimens were then fixed in 2.5% glutaraldehyde for 30 minutes, washed in Sorensen’s phosphate buffer, and then submitted to secondary fixation in 1–2% osmium tetroxide for 1 h. The fixed specimens were dehydrated in graded ethyl alcohols, and then embedded in molds by infiltration of graded embedding medium in propylene oxide. The hardened specimens were cut into ultra-thin sections (60–90 nM) with a UCT7 ultratome (Leica, Wetzlar, Germany). The ultra-thin sections were stained with uranyl acetate and lead citrate. Isolation membrane and autophagosome formation were observed by transmission electron microscopy (TEM) under an H-7700 microscope (Hitachi, Tokyo, Japan).

### 4.11. Data Analysis

Group data are displayed as means ± standard errors of the mean (SEMs). Differences between groups were calculated by Student’s unpaired two-tailed *t-*tests or one-way analyses of variances (ANOVAs) followed by Tukey’s post hoc tests. A *p* < 0.05 was considered statistically significant.

## 5. Conclusions

In conclusion, we reported herein for the first time that FGF21 regulates pancreatic islet autophagy through a mechanism, probably involving the suppression of AMPK-mTOR signaling. The present study provides new insight into the protective mechanism of FGF21 against glucolipotoxicity-induced islet dysfunction in obesity-associated T2DM. In addition to providing empirical evidence supporting the novel, physiological roles of FGF21 in pancreatic islets, this study posits that FGF21 may act as an efficacious anti-diabetic agent in clinical settings.

## Figures and Tables

**Figure 1 ijms-20-02517-f001:**
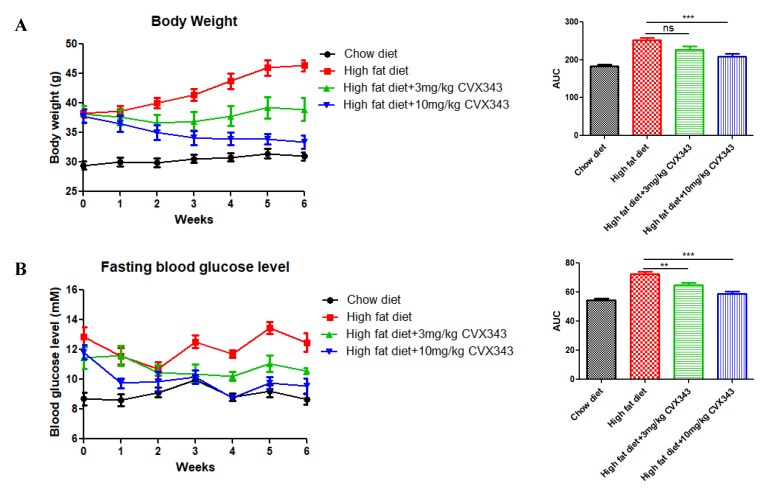
Effects of the fibroblast growth factor 21 (FGF21) analog CVX343 on body weight and glucose homeostasis of mice with high-fat diet (HFD)-induced type 2 diabetes mellitus (T2DM). (**A**) Weekly mean body weight (BW) of each group and comparison of BW group with AUCs. (**B**) Weekly fasting blood glucose levels for each group and comparison of fasting blood glucose with AUCs. (**C**) Glucose tolerance results measured at 15-min intervals over a 2-h IPGTT and comparison of group IPGTT with areas under the curve (AUCs). (**D**) Blood glucose levels measured at 15-min intervals over a 2-h ITT and comparison of group ITT with AUCs. Data are means ± SEMs; *n* = 5–10 per group; ns, non-significant; * *p* < 0.05, ** *p* < 0.01, and *** *p* < 0.001.

**Figure 2 ijms-20-02517-f002:**
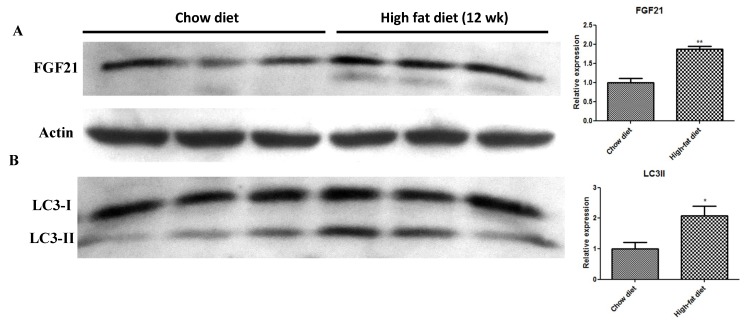
Effects of HFD feeding and high-glucose/high-PA conditions on the protein expression of FGF21 and LC3-II in pancreatic islets. Western blot images and analyses for the expression of FGF21 (**A**) and LC3-II (**B**) in islets isolated from mice fed with a HFD for 12 weeks. (**C**) Western blot analysis of LC3-II expression in isolated islets during high-glucose/high-PA treatment. (**D**) Western blot analysis of FGF21 expression in isolated islets after a 24-h high-glucose/high-PA treatment. Data are means ± SEMs; *n* = 3–6 per group; * *p* < 0.05, and ** *p* < 0.01.

**Figure 3 ijms-20-02517-f003:**
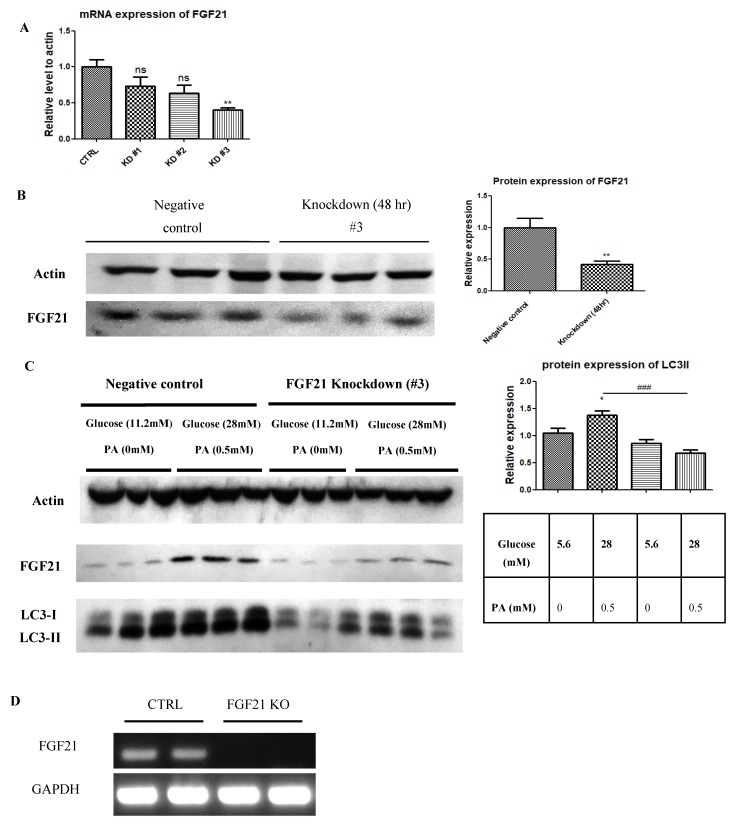
Identification of LC3-II expression in *FGF21* KO mice and INS-1E cells with *FGF21* knockdown. Comparison of knockdown efficiency achieved by three anti-*FGF21* siRNAs in INS-1E cells (*n* = 6 replicates/group). The siRNA #3 proved to be the most effective tested siRNA for knocking down; FGF21 mRNA transcription detected by real time PCR (**A**) and FGF21 protein expression detected by Western blot (**B**) (*n* = 6 replicates/group). (**C**) Suppression of LC3-II protein expression (Western blot) in anti-*FGF21* siRNA #3 knocked down INS-1E β-cells, compared to cells exposed to control siRNA, following a treatment with 24-h high-glucose/high-PA exposure (*n* = 6 colonies/group). (**D**) RT-PCR analysis for the mRNA expression of FGF21 in pancreatic islets of *FGF21* KO mouse. (**E**) Suppression of LC3-II protein expression (Western blot) in *FGF21* KO islets, compared to WT islets, following 24-h high-glucose/high-PA treatment (*n* = 6 islets/group). # Also, * *p* < 0.05, ** *p* < 0.01 and ### *p* < 0.001.

**Figure 4 ijms-20-02517-f004:**
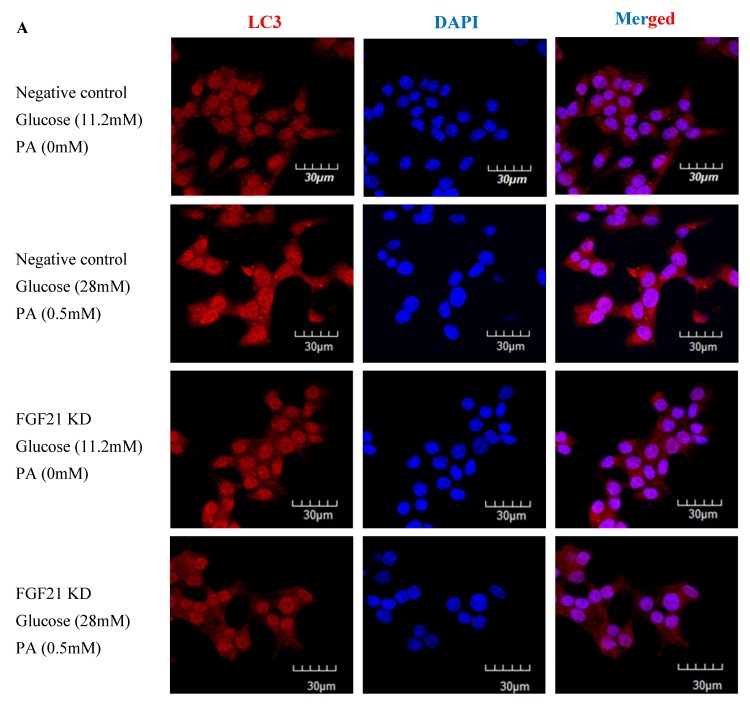
Induction of reduced formation of isolation membranes and autophagosomes under high-glucose/high-PA conditions in INS-1E cells with *FGF21* knockdown and isolated islets with *FGF21* KO mice. (**A**) Fluorescent immunohistochemistry showing decreased expression of cytoplasmic LC3-II in high-glucose/high-PA-treated INS-1E cells compared to non-treated control cells. (**B**) TEM images demonstrating reduced formation of isolation membranes (blue arrow) and autophagosomes (red arrow) in *FGF21* KO islets under diabetic conditions. The yellow square is the area that was magnified. Quantitative summaries of these TEM observations of isolation membranes (**C**) and autophagosomes (**D**). Data are means ± SEMs; *n* = 8–9; ## *p* < 0.01, and *** *p* < 0.001.

**Figure 5 ijms-20-02517-f005:**
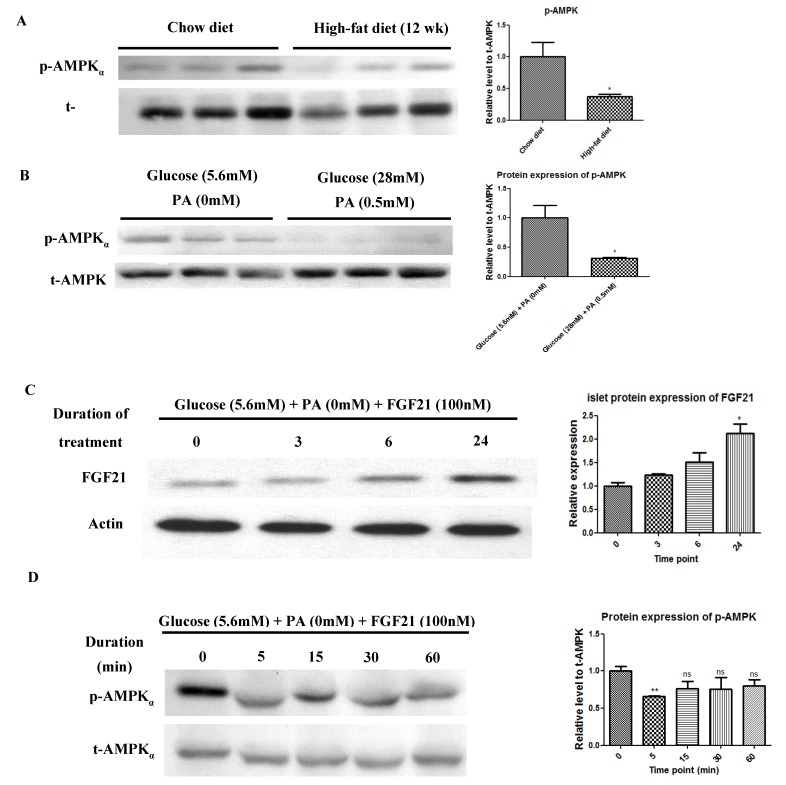
Reduction of AMPK phosphorylation in isolated pancreatic islets and INS-1E cells by HFD feeding and ex vivo high-glucose/high-PA treatment. (**A**) Western blot analysis of AMPK in islets isolated from mice fed with a HFD or chow diet for 12 weeks. (**B**) Western blot analysis of AMPK in isolated islets treated with the presence or absence of high-glucose/high PA for 24 h. Increased FGF21 expression and reduced AMPK phosphorylation in isolated pancreatic islets by exogenous administration of FGF21 recombinant protein. (**C**) Western blot analysis of FGF21 in islets treated with FGF21 recombinant protein for indicated durations. (**D**) Western blot analysis of AMPK in islets treated with FGF21 recombinant protein for indicated durations. All data are means ± SEMs; *n* = 5–6 and * *p* < 0.05, ** *p* < 0.01.

**Figure 6 ijms-20-02517-f006:**
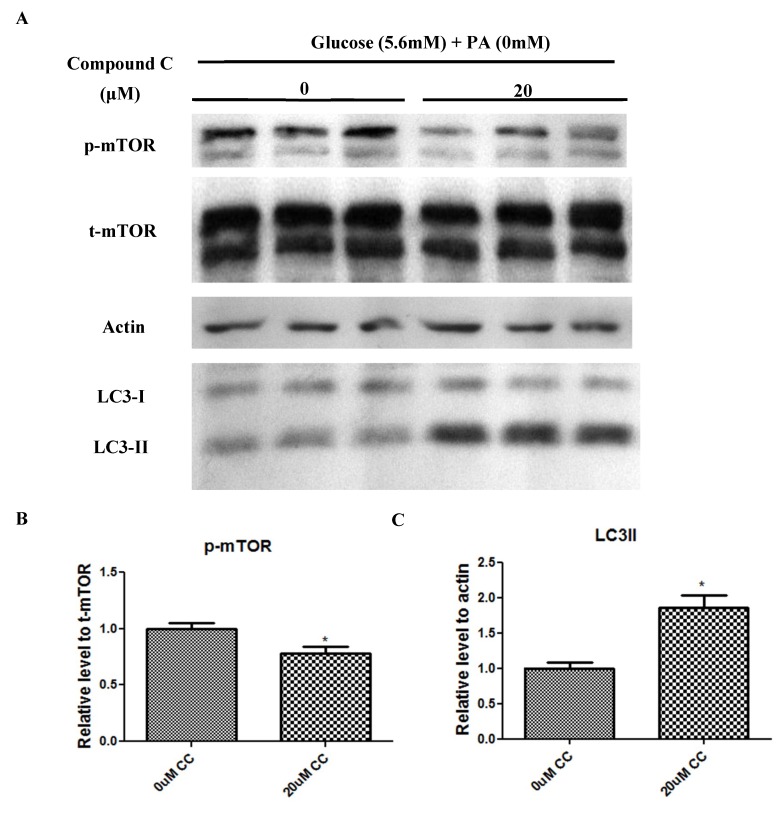
The AMPK blocker compound C reduced the phosphorylation of mTOR while increasing the expression of LC3-II protein in isolated pancreatic islets. Western blots and quantitative analyses for the protein levels of mTOR (**A**,**B**) and LC3-II (**A**,**C**) in isolated islets treated, or not, with 20 μM compound C for 24 h. Data are means ± SEMs; *n* = 6 and * *p* < 0.05. vs. the non-treated control group.

**Table 1 ijms-20-02517-t001:** Real-time PCR primer sequences and siRNA sequences.

Gene	Forward (5′ → 3′)	Reverse (5′ → 3′)
Mouse *GAPDH*	GCACAGTCAAGGCCGAGAAT	GCCTTCTCCATGGTGGTGAA
Mouse *FGF21*	CQTCTGCCTCAGAAGGACTC	AAGGCTCTACCATGCTCAGG
Rat *β-actin*	TTTAATGTCACGCACGATTTC	CCCATCTATGAGGGTTACGC
Rat *FGF21*	AGATCAGGGAGGACGGAACA	TCAGGATCAAAGTGAGGCGAT
Negative control siRNA	UUCUCCGAACGUGUCACGUTT; ACGUGACACGUUCGGAGAATT	
Rat siRNA-*FGF21* (#1)	CAACCAGAUGGAACUCUCUAUGGAU; AUCCAUAGAGAGUUCCAUCUGGUUG	
Rat siRNA-*FGF21* (#2)	GCAGUUUCAGAGAGCUGCUGCUUAA; UUAAGCAGCAGCUCUCUGAAACUGC	
Rat siRNA-*FGF21* (#3)	CCCUGAGCAUGGUAGAGCCUUUGCA; UGCAAAGGCUCUACCAUGCUCAGGG	

**Table 2 ijms-20-02517-t002:** Antibodies used in Western blotting and immunohistochemistry.

Antibody	Dilution	Host Species	Supplier
FGF21	1:1000	Rabbit	Abcam
LC3	1:1000	Rabbit	Novus
p-AMPK	1:1000	Rabbit	Cell Signaling
t-AMPK	1:1000	Rabbit	Cell Signaling
β-actin	1:1000	Mouse	Santa Cruz
p-mTOR	1:1000	Rabbit	Cell Signaling
t-mTOR	1:1000	Rabbit	Cell Signaling
HRP-anti-rabbit IgG	1:1000	Donkey	Amersham
HRP-anti-mouse IgG	1:1000	Sheep	GE Healthcare
Alexa Fluor^®^ 568 anti-rabbit IgG	1:2000	Donkey	Life Technologies

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
