# Peer review of "Fibroblast Growth Factor 21 Stimulates Pancreatic Islet Autophagy via Inhibition of AMPK-mTOR Signaling"

_ijms, 2019, doi:10.3390/ijms20102517_

Round 1

Reviewer 1 Report

This is an important high-quality straight-forward paper needing only minor revison.

1) Abstract: Change "compound C" to "AMPK inhibitor compound C".

2) Abstract and main text: If treatment effects of FGF21 were reported using its analog CVX343, change "FGF21" to "FGF21 analog CVX343".

3) Line 113 (subheading): change "CVX343" to "FGF21 analog CVX343".

4) Were the morphological analyses done blinded?

Author Response

See attached pdf file.

Reviewer 2 Report

The authors describe in this work the effects of FGF21 on pancreatic islet autophagy. The authors show that islet protein expression of FGF21 was induced after HFD treatment correlated with increased LCII. The manuscript reports potentially interesting data. However, the amount of new data presented is limited and the conclusion that glucolipotoxicity-induced FGF21 activation mediates islet autophagy via AMPK inhibition is not fully confirmed by data presented.

Specific comments: 

- The CVX343 should be introduced as FGF21 analogue before experiments description (currently, the first mention of what is CVX343 is in the title of the figure and then in the materials and methods section). 

- Figure 1A: why the starting weight for the normal chow and HFD is so different. When were this animals switched on HFD? If earlier, what is the meaning of week 0 / week 1? what is your starting point in this case? 

- Figure 1: I was looking for the raw data, with individual values for each mouse, but I didn’t find them. It would be preferable to have access to the data file on which the graphs in figure 1 were generated. My questions relates mostly to low standard deviation in glycemia graph, especially with 10 mice. How long were the mice fasted (there is no indication in methods part). Also, would be preferably to show the individual AUC in the bar graphs (scatter plots with bars). 

- Is there a statistical difference between the chow diet and the high-dose CVX343 (10 mg/kg BW)? From the graphs seem quite similar and would be a more interesting conclusion to compare also with chow diet. 

- Figure 2A: what is the second band appearing in the HFD samples on FGF21 blot?

- No protein ladders were used in the manuscript! 

- Figure 4A: Not sure what I should see in this figure. The LC3 staining is usually presenting a highly punctuated pattern labelling the autophagosomal membranes. In the images presented no clear punctuated staining is present in any of the conditions showed. There is a slight difference in the intensity staining, on which the authors base their conclusion of decreased cytoplasmic LC3-II expression in high-glucose/high-PA-treated INS-1E cells with FGF21 knockdown, compared to non-treated control cells. However, very few can be concluded using these images. A higher magnification would help. Also, a quantification of the auphagosomes per cells might also be useful for reaching this conclusion. Especially that the TEM images were performed on islets while IF was done on Ins-1E cells. How is LC3 staining in FGF21 KO islets? 

- There is an entire paragraph on circulating FGF21 in the discussion however there is no result presented in the paper on the circulated FGF21 in the HFD.  

Author Response

See attached pdf file.

Round 2

Reviewer 2 Report

The authors addressed all the criticisms raised by this reviewer, but somewhat insufficient. I still think that the paper might gain some value is better imunofluorescence images presenting the autophagosomal membranes staining in a zoomed magnification.

However, these findings are very interesting and potentially important.